# Impact of Cold Stress on Leaf Structure, Photosynthesis, and Metabolites in *Camellia weiningensis* and *C. oleifera* Seedlings

**Hongyun Xu [1], Chengling Huang [1], Xian Jiang [2], Jing Zhu [2], Xiaoye Gao [1],* and Cun Yu [2],***

[1] College of Eco-Environmental Engineering, Guizhou Minzu University, Guiyang 550025, China; xuhongyun@gzmu.edu.cn (H.X.); hcling@gzmu.edu.cn (C.H.)
[2] College of Forestry, Guizhou University, Guiyang 550025, China; gs.xjiang19@gzu.edu.cn (X.J.); gs.jingzhu19@gzu.edu.cn (J.Z.)
* Correspondence: gaoxiaoye@gzmu.edu.cn (X.G.); cyu@gzu.edu.cn (C.Y.)

**Abstract:** *Camellia weiningensis* Y. K. Li. sp. *nov.* (CW) is an endemic oil-tea species in Guizhou province, distributed in the alpine karst area, which exhibits cold resistance and better economic characters than *C. oleifera* (CO). The mechanism of cold response in CW seedlings has not been studied in depth. Herein, we performed anatomical, physiological, and metabolic analyses to assess the impact of cold stress on leaf structure, photosynthesis, and metabolites in CW and CO seedlings. Anatomical analysis of leaves showed CW seedlings had greater leaf and palisade thicknesses, tissue structure tightness, and palisade-spongy tissue ratio to enhance chilling stress (4 °C) tolerance, but freezing stress (−4 °C) caused loosening of the leaf tissue structure in both CW and CO seedlings. Photosynthetic analysis showed a reduction in the chlorophyll (Chl) fluorescence ($F_v/F_m$) and photosynthetic parameters under freezing stress in both CW and CO seedlings. Cold stress increased the abscisic acid (ABA) contents in both the *Camellia* species, and CW exhibited the highest ABA content under −4 °C treatment. Additionally, the indole-3-acetic acid (IAA) content was also increased in CW in response to cold stress. An obviously distinct metabolite composition was observed for CW and CO under different temperatures, and significantly changed metabolites (SCMs) were enriched under freezing stress. Prenol lipids, organooxygen compounds, and fatty acyls were the main metabolites in the two *Camellia* species in response to cold stress. The top key SCMs, such as medicoside G, cynarasaponin F, yuccoside C, and methionyl-proline were downregulated under freezing stress in both CW and CO. The contents of some key metabolites associated with sugar metabolism, such as UDP-glucose, UDP-D-apiose, and fructose 6-phosphate, were higher in CW than in CO, which may contribute to enhancing the cold resistance in CW. Our findings are helpful in explaining how CW adapt to alpine karst cold environments, and will provide a reference for cold tolerance improvement and application of stress-resistant breeding of *Camellia* in alpine and cold areas.

**Keywords:** cold stress; leaf structure; photosynthesis; metabolites; *Camellia weiningensis*; *Camellia oleifera*

## 1. Introduction

The oil-tea tree (*Camellia oleifera* Abel.) (CO) is an economic oil plant, widely distributed in southern China [1]. The tea oil contains abundant oleic acid, which has high nutritive value and health-promoting functions [2]. The habitat of tea oil trees is characterized by extremely low temperatures in winter and frost in early spring, which often cause cold injury to seedlings and restrict the development of *C. oleifera* industry [3].

*Camellia weiningensis* Y. K. Li. sp. *nov.* (CW) is an important species in the *camellia* genus, which is an endemic economic woody oil tree species, specifically distributed in Weining County, Guizhou Province, China [4]. CW is suitable for distribution in high altitude areas and exhibits strong cold resistance. The mean air temperature in the distribution area of CW can reach 1.6 °C in January [4]. However, the physiological, anatomical, and

metabolic mechanisms involved in the response of CW seedlings to cold stress environment need to be explored in greater depth.

To adapt to various environments, plants develop a number of specific anatomical structures [5]. The structural features of leaves, which are the main organs involved in photosynthesis and transpiration, are easily affected by abiotic stress [6]. Plants with strong cold resistance have higher leaf cell tense ratio (CTR) and lower spongy ratio (SR) [5]. Compared to the 28 °C treatment, 8 °C treatment significantly reduces the CTR and increases the SR of tung tree seedlings [7]. The main indices associated with cold tolerance in CO are upper epidermis thickness, sponge tissue thickness, tissue structure tightness, and stoma density [8]. Leaf thickness in two *Camellia* cultivars (Hua Shuo and Hua Xin) increases under 6 °C treatment [9]. However, the leaf anatomical structure of CW seedling leaves adapted to cold stress has not been studied.

Low temperature injury causes photoinhibition because of reduction in stomatal conductance or photochemical activity [10]. The activities of photosystem (PS) I and II are inhibited in the leaves of CO under sub-chilling stress [11]. The photosynthetic parameters of *C. oleifera* cultivars "Huashuo" and "Huaxin" are evidently decreased under low temperature treatment [12]. In CW seedlings, many genes associated with photosynthetic systems are down-regulated, and Chl content and $P_n$ index are decreased under cold stress [13]. It is necessary to investigate cold resistance by comparing photosynthetic indices in CW and CO seedlings.

Plants can avoid low temperature injury by undergoing metabolic changes. In over-wintering plants, the most common metabolites to combat cold stress are betaines, sugars, polyols, polyamines, and amino acids [14]. Exogenous ALA enhanced *C. sinensis* cold resistance by regulating soluble carbohydrates and flavonoids metabolites [15]. Previous study exhibited the metabolic changes of CO when suffered from abiotic stresses. For instance, key metabolites are induced in CO under drought stress, such as amino acids, carbohydrates and phenols [16]; mulching treatment in CO results in changes in the contents of tyrosine, tryptophan, and several flavonoids and polyphenol metabolites to alleviate drought stress [17]. Different altitudes and cold stresses also have various effects on fatty acids and phytohormones synthesis, for instance, Rahman et al. [18] found that ABA contents enhance as temperature decreased and altitude increased, but IAA shows an opposite response; Tsegay et al. [19] found that the fatty acids levels of the coffee plants reduce with increasing altitude; eleven fatty acids are identified in chickpea, whose unsaturated fatty acids ratio enhances during cold acclimation phase [20]. It is necessary to understand the changes in metabolism of alpine CW under low temperature environment.

CW has an excellent economic value, and is suitable for planting in alpine karst areas. We hypothesized that cold stress had an important effect on leaf morphology, photosynthesis, and metabolites in CW seedlings. In this study, using CO, grown widely in Guizhou province as a reference, we performed a comprehensive comparative analysis of CW with the following objectives: (1) To analyze differences in the leaf morphology of the two *Camellia* species under cold stress; (2) to investigate the effects of cold stress on photosynthesis; (3) to identify the key metabolites associated with cold resistance. Our results found that cold stress caused leaf structure changes, and photosynthesis inhibition in CO and CW seedlings, some key SCMS and sugar metabolism pathways were important for CW in response to cold stress. Our study provides a more detailed understanding of the cold resistance mechanism of CW, and lay the foundation for the cultivation of CW in cold karst areas.

## 2. Materials and Methods

### 2.1. Plant Materials and Treatments

*Camellia oleifera* (Chang Lin) and *C. weiningensis* seedlings were obtained from Yuping Dong Autonomous County (27°31′47″ N, 108°93′01″ E) and Weining County (27°19′57″ N, 104°13′41″ E) in Guizhou Province, China, respectively. One-year-old healthy seedlings were chosen and replanted in plastic pots (diameter, 12 cm; height, 15 cm). The plots were

filled with a mixture of peat soil, loess, and perlite at a proportion of 2:1:1. The seedlings were cultivated in an artificial climate incubator in Guizhou university under conditions of $20 \pm 2\,°C$ with 12/12 h (light/dark) cycles, a relative air humidity of 65–70% and light intensity of $400\,\mathrm{\mu mol\,m^{-2}\,s^{-1}}$. After 3 months, the uniformly sized healthy seedlings were exposed to different temperatures as follows: CW set to $20\,°C$ for 24 h (A), CW set to $4\,°C$ for 24 h (B), CW set to $-4\,°C$ for 24 h (C), CO set to $20\,°C$ for 24 h (D), CO set to $4\,°C$ for 24 h (E), and CO set to $-4\,°C$ for 24 h (F). For each treatment, at least 30 seedlings were chosen.

### 2.2. Studies on the Anatomical Structure of Leaves

The anatomy of leaves from seedlings subjected to different temperature treatments for 24 h was observed. The mature leaves were selected and cut into $5\,\mathrm{mm} \times 5\,\mathrm{mm}$ pieces, and then soaked in the 38% formaldehyde, 70% alcohol, and acetic acid solution (5:90:5, $v/v/v$) for 24 h. The samples were then dehydrated in a graded ethanol series [9], embedded in paraffin, and sectioned with a microtome (Leica RM2016, Wetzlar, Germany). The sections were stained with the double dyes (Safranine and Fast green) according to Ruzin's method [21], and imaged under a microscope (NIKON ECLIPSE E100, Tokyo, Japan). The thicknesses of leaf, upper and lower epidermis, and spongy and palisade tissues were measured using the Image J software (version 1.8.0) [1]. The palisade tissue-spongy tissue ratio (=palisade/spongy tissue thickness), tissue structure tightness (=[palisade tissue/leaf thickness] $\times$ 100%), and tissue structure looseness (=[spongy tissue/leaf thickness] $\times$ 100%) were calculated [1].

### 2.3. Measurement of Photosynthetic Parameters

Net photosynthetic rate, transpiration rate, and stomatal conductance were determined in mature leaves, after 1 d recovery from cold stress [11], on a sunny morning using a Li-6800xt (LI-COR, Lincoln, NE, USA). The gas exchange parameters were set as described by Xu and Yu [13]. Water use efficiency was calculated as the proportion of net photosynthetic rate to transpiration rate.

### 2.4. Measurement of Chlorophyll Fluorescence and Chlorophyll Content

In order to be consistent with photosynthetic parameters, chlorophyll fluorescence and chlorophyll content of leaves were determined after 1 d recovery from cold stress. The initial fluorescence ($F_0$) and the maximum fluorescence ($F_m$) were determined after 30 min of dark adaptation using a portable chlorophyll fluorometer (Cl-340, WALZ Company, Wurzburg, Germany). The $F_v/F_m$ was determined using a previously described equation as $(F_m - F_0)/F_m$ [22]. The chlorophyll contents of CW and CO leaves were determined after 80% ($v/v$) acetone extraction, as described by Lichtenthaler and Wellburn [23].

### 2.5. Quantitation of Plant Hormones

Fresh leaves were used for determining the content of plant hormones after cold stress treatment. Leaves are ground into a powder with liquid nitrogen, and the leaf powder (0.1 g per sample) was added to 9 mL phosphate buffer (pH 7.4). The mixture was then centrifuged for 30 min at $4\,°C$ and the supernatant was collected and used to measure the content of gibberellin (GA), jasmonic acid (JA), abscisic acid (ABA), and 3-indoleacetic acid (IAA) using enzyme-linked immunosorbent assay (ELISA) kits (Quanzhou Ruixin Biological Technology Co., Ltd., Quanzhou, China). According to the instruction, 50 μL of the supernatant and biotinylated antibodies were added to the 96-well microtiter trays, respectively, then incubated at $37\,°C$ for 60 min. Subsequently, the microplate was washed, and the liquid in the wells was discarded, then each well was filled with washing liquid, and spin-dried after allowing to stand for 10 s, and this step was repeated three times. A total of 50 μL enzyme-labeled avidin was added to the wells, and incubated at $37\,°C$ for 30 min; then the microplate were washed again. Finally, 50 μL chromophoric reagen A and B was added to the wells, then reacted at $37\,°C$ for 15 min after shaking and mixing, and 50 μL termination solution was added to each well, then the microtiter trays were read

with the microplate reader (JC-1086A, Qingdao, China) at a wavelength of 450 nm. All samples were repeated three times.

### 2.6. Untargeted Metabolomic Analysis

Metabolite extraction: Fifty milligram crushed leaf samples were accurately weighed in 2 mL centrifuge tubes and extracted with 400 µL methanol:water (4:1, *v/v*); 0.3 mg/mL L-2-chlorophenylalanine solution was added as an internal standard. The mixture was ground using a Wonbio-96c high-throughput tissue crusher (Wanbo Biotechnology, Shanghai, China) at −10 °C at 50 Hz for 6 min, and then ultrasonicated at 40 KHz for 30 min in an ice-bath. The extract was incubated at −20 °C for 30 min and centrifuged at 13,000 rpm at 4 °C for 15 min. The supernatant was filtered by 0.22 µm filters, then used for LC-MS analysis. Additionally, quality control (QC) sample was the mixture of all samples, which was used to determine the stability of the system.

LC-MS analysis: the LC-MS analysis was performed on a Thermo UHPLC-Q Exactive HF-X system using an ACQUITY UPLC HSS T3 column (100 mm × 2.1 mm i.d., 1.8 µm; Waters, Milford, MA, USA). A 2 µL aliquot of the sample was injected into the column held at 40 °C. The mobile phase A consisted of 95% water, 5% acetonitrile, and 0.1% formic acid; the mobile phase B consisted of 47.5% acetonitrile, 47.5% isopropanol, 5% water, and 0.1% formic acid. The proportion of mobile phase A to B was changed with the following gradient elution process: 0–3.5 min, 100% A, flow rate of 0.4 mL/min; 3.5–5 min, 75.5% A and 25.5% B, flow rate of 0.4 mL/min; 5–5.5 min, 35% A and 65% B, flow rate of 0.4 mL/min; 5.5–7.4 min, 100% B, flow rate of 0.4 mL/min; 7.4–7.6 min, 100% B, flow rate of 0.6 mL/min; 7.6–7.8 min, 48.5% A and 51.5% B, flow rate of 0.6 mL/min; 7.8–9 min, 100% A, flow rate of 0.5 mL/min; 9–10 min, 100% A, flow rate of 0.4 mL/min. Mass spectrometry (MS) analysis was performed using a mass spectrometer (Q-Exactive HF-X, Thermo Scientific, Waltham, MA, USA) equipped with an electrospray ionization (ESI) source operating in positive (pos) and negative (neg) modes. The optimal parameters were set as Zhang et al. [24].

### 2.7. Separation of Metabolites and Data Analysis

The raw data were processed using the Progenesis QI software (Waters Corporation, Milford, MA, USA) picking and identification of peaks. The data matrix contained mass-to-charge ratio (*m/z*) values, the retention time (RT), and peak intensity. Peak area is the mass spectrum response intensity of metabolites, which indicates the abundance (content) of metabolites. Peak areas were normalized to the corresponding peak areas for the QC samples. Mass spectra of these metabolic features were identified using the accurate masses, MS/MS fragment spectra, and differences in the isotope ratio, with searches in internal and public databases. The variable importance of the projection (VIP) score generated from orthogonal partial least squares discriminate analysis (OPLS-DA) was used to determine the best differentiated. Metabolites with VIP values from the OPLS-DA model ≥1.0, and *p* value < 0.05 were defined as significantly changed metabolites (SCMs) [25]. The metabolites were annotated based on the Human Metabolome Database (HMDB) (http://www.hmdb.ca/; accessed on 22 February 2022) and the Kyoto Encyclopedia of Genes and Genomes (KEGG) database (https://www.genome.jp/kegg/pathway.html; accessed on 15 March 2022). Metabolites were subjected to principal component analysis (PCA) and hierarchical cluster analysis (HCA) based on the Majorbio Cloud Platform. To compare the metabolic differences in each *Camellia* species at different temperatures, the B vs. A and C vs. A groups (A was the control) were set as chilling and freezing stress for CW, respectively; the E vs. D and F vs. D groups (D was the control) were set as chilling and freezing stress for CO, respectively. To compare the metabolic differences between the two species under different temperature, the group ABC vs. DEF was analyzed.

### 2.8. Statistical Analysis

The parameters associated with leaf anatomy, photosynthesis, and plant hormones were compared using one-way analysis of variance (ANOVA) with the comparison of mean

values performed using Duncan's test at $p < 0.05$. Data marked with different letters are statistically significant. Two-way ANOVA was used to determine the statistical significance ($\alpha = 0.05$) of temperatures and species on leaf anatomy, photosynthesis, as well as plant hormone contents.

## 3. Results

### 3.1. Changes in Anatomical Features of Leaves in the Two Camellia Species under Cold Stress

As shown in Figure 1, leaf anatomical structure were changed under different temperature treatment through quantitative data available from image analysis. Under normal conditions, the thickness of CO leaves was 31.56% higher than that of CW leaves (Figure 1b). Compared with the 20 °C treatment, at 4 °C, the thickness of leaves, upper epidermis, lower epidermis, and palisade in CW seedlings increased significantly; these indices decreased significantly in CO seedlings at 4 °C (Figure 1b–e). The tissue structure tightness in CW increased at 4 °C compared with that at 20 °C (Figure 1h). Under −4 °C treatment, the thickness of leaf, palisade, and spongy tissue, palisade-spongy tissue ratio, and tissue structure tightness were significantly reduced both in CO and CW compared with that at 20 °C (Figure 1b,e–h). Meanwhile, Two-way ANOVAs displayed that leaf anatomical structure were significantly affected by temperature and species ($p < 0.05$, Table S1).

### 3.2. Cold Stress Reduces $F_v/F_m$ Value and Photosynthesis

$F_v/F_m$ parameter is certainly the most used Chl fluorescence parameter. As shown in Figure 2a, there was no significant difference in the $F_v/F_m$ value for CW between the 20 and 4 °C treatments, but the ratio declined significantly at −4 °C. Compared with the 20 °C treatment, the $F_v/F_m$ ratio for CO decreased significantly under 4 and −4 °C conditions. Cold stress inhibited photosynthesis in CW and CO seedlings. The photosynthetic parameters, such as net photosynthetic rate, stomatal conductance, transpiration rate, water use efficiency, and chlorophyll content, were reduced under 4 and −4 °C conditions (Figure 2b–f). Meanwhile, two-way ANOVAs showed that different temperature treatments had significant effects on photosynthetic parameters ($p < 0.05$, Table S2), but there was no significant difference in $F_v/F_m$, stomatal conductance, transpiration rate, and Chl content between two *Camellia* species ($p > 0.05$, Table S2).

### 3.3. Quantitation of Plant Hormones

The contents of GA and JA in CW seedlings exposed to −4 and 4 °C were lower than in the 20 °C treatment (Figure 3a,c), but the content of ABA and IAA increased significantly compared with that in the control (20 °C) (Figure 3b,d). Chilling and freezing stress also significantly increased the ABA content in CO by 18.42% and 8.04%, respectively ($p < 0.05$, Figure 3b). The JA contents in CO in the cold conditions were significantly lower than in the 20 °C conditions (Figure 3c). For CO seedlings, the IAA content declined significantly with decreasing temperature (Figure 3d). Two-way ANOVAs displayed that GA, JA, and IAA contents were significantly affected by temperature and species ($p < 0.05$, Table S3), but ABA content showed significant difference between species ($p > 0.05$, Table S3).

### 3.4. Overview of Metabolite Profiling

There are 1016 metabolites identified in all the samples, including 507 identified in the positive mode and 509 in the negative mode. PCA was used to investigate the difference among all the groups (Figure 4a,b). The two *Camellia* species had a distinct metabolite composition under different temperature conditions. For CW or CO seedlings, treatments C and F could be obviously separated with other samples, but some samples in the 4 °C treatment (B or E) could not be separated with samples in the 20 °C treatment (A or D), suggesting that freezing stress caused greater differences in metabolites. The HCA was conducted to determine the accumulation patterns of metabolites in the different samples (Figure 4c). The heat map displays that the expression patterns of metabolites were different

between CW and CO. For each *Camellia* species, the metabolite contents changed under different temperature stresses.

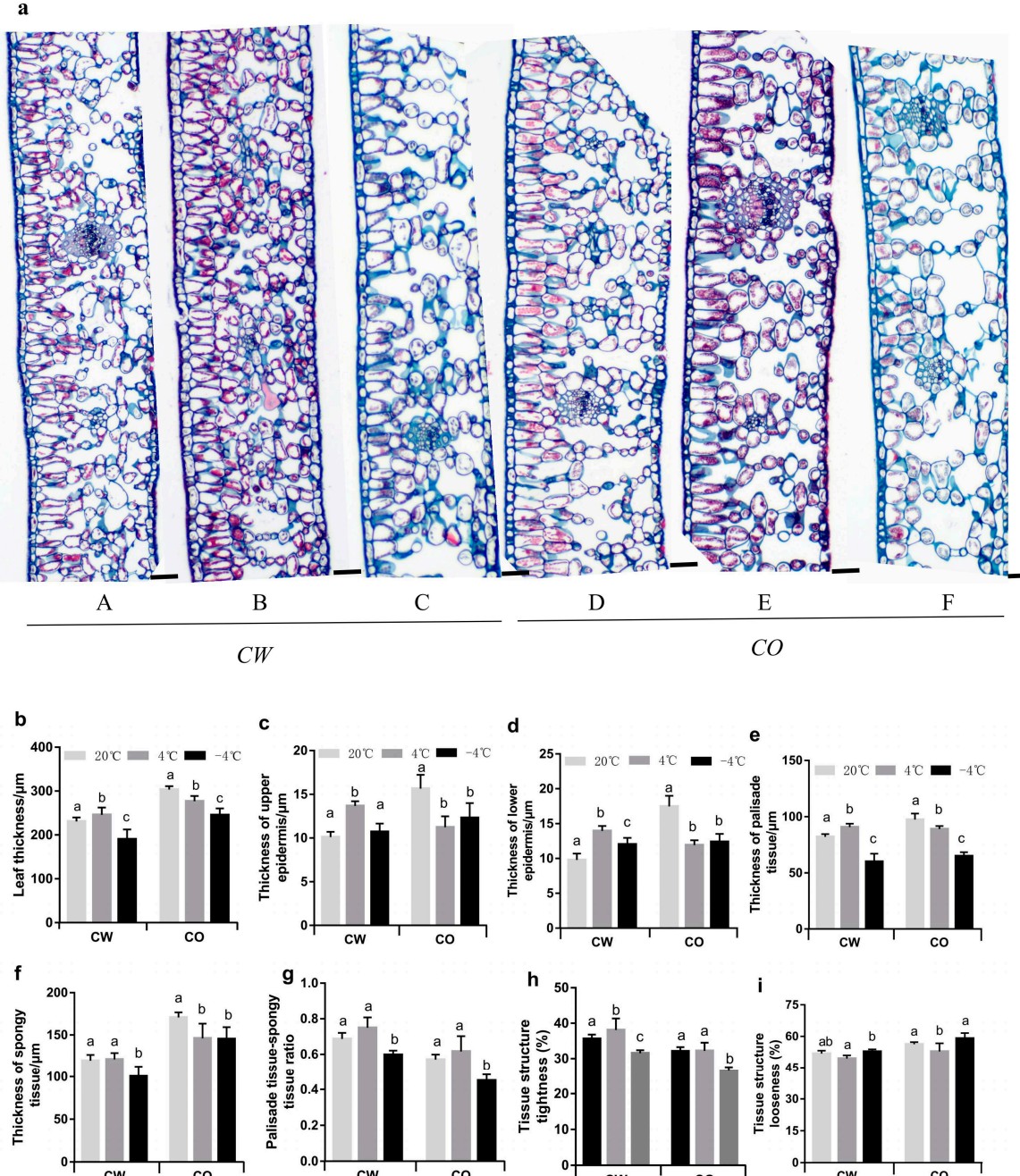

**Figure 1.** Effects of different temperatures on leaf anatomical structure of two Camellia species. (**a**) The representative pictures of *C. weiningensis* (CW) and *C. oleifera* (CO) seedlings, Scale = 50 μm; A, CW treated with 20 °C; B, CW treated with 4 °C; C, CW treated with −4 °C; D, CO treated with 20 °C; E, CO treated with 4 °C; F, CO treated with −4 °C. (**b**–**i**) Determination of leaf anatomic structure indexes, (**b**) leaf thickness, (**c**) thickness of upper epidermis, (**d**) thickness of lower epidermis, (**e**) thickness of palisade tissue, (**f**) thickness of spongy tissue, (**g**) palisade tissue-spongy tissue ratio, (**h**) tissue structure tightness, (**i**) tissue structure looseness. Six plants of each of the two species from cold treatment and control were used for the measurements, five images per plant were taken. Different letters in each species indicate significant differences among different temperature treatments ($p < 0.05$).

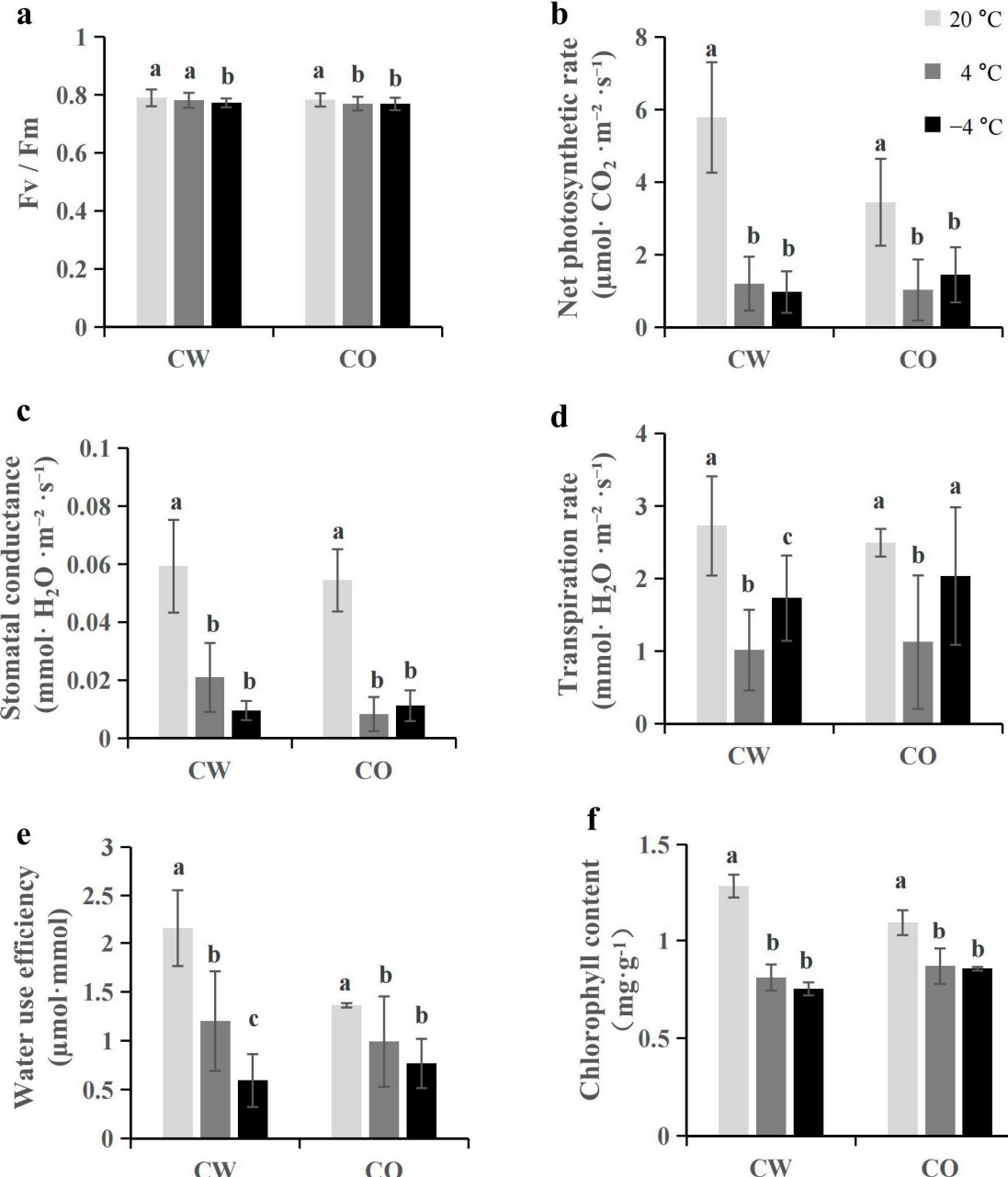

**Figure 2.** Effects of cold stress on photosynthesis in two *Camellia* species. (**a**) $F_v/F_m$, (**b**) net photo-synthetic rate, (**c**) stomatal conductance, (**d**) transpiration rate, (**e**) water use efficiency, (**f**) chlorophyll content. CW, *Camellia weiningensis*; CO, *Camellia oleifera*. CW and CO seedlings were exposed to 20 °C, 4 °C (chilling stress) and −4 °C (freezing temperature) for 24 h, then photosynthetic parameters were measured after 1 d recovery from cold stress. Nine plants per treatment were used as one replicate, four leaves per plant were used for measurements, and the experiment was set up for three repetitions. Values (means ± SE) followed by different letters in each species indicate significant differences among treatments ($p < 0.05$).

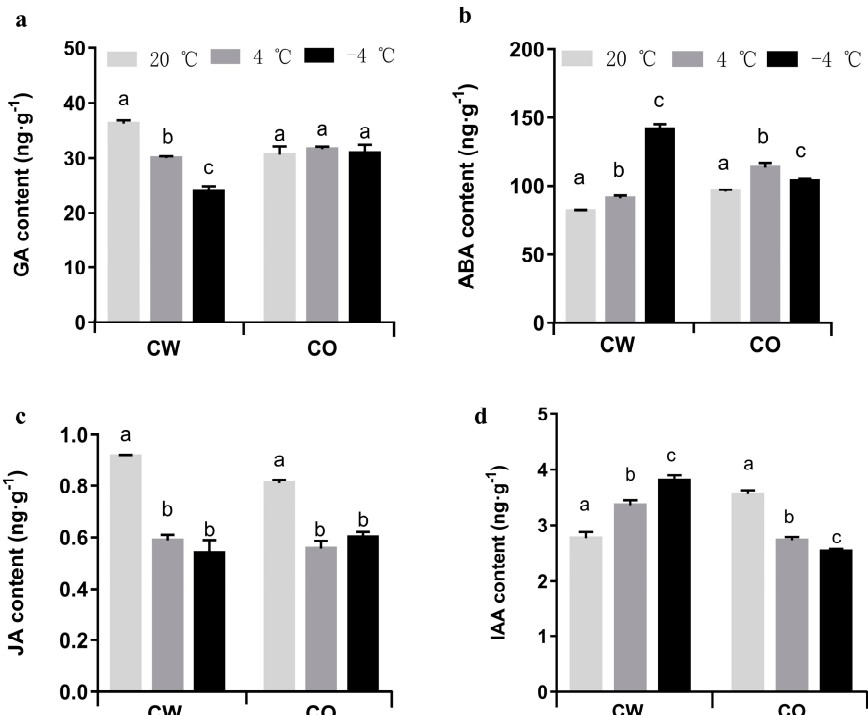

**Figure 3.** Effect of low temperature on the contents of endogenous hormone contents. (**a**) gibberellin (GA), (**b**) abscisic acid (ABA), (**c**) jasmonic acid (JA), (**d**) 3-indoleacetic acid (IAA). CW, *Camellia weiningensis*; CO, *Camellia oleifera*. The leaves of nine plants mixed as a repetition, three repetitions were taken. Data represent means ± SE ($n = 3$). Bars not sharing the same lowercase letter in each species are significantly different among treatments according to Duncan's multiple range test ($p < 0.05$).

*3.5. Significantly Changed Metabolites under Cold Stress*

A total of 302 and 172 SCMs were identified in CW and CO under cold stress, respectively, suggesting that more SCMs were induced by chilling and freezing stress in CW (Figure S1a). There were 126 SCMs in B vs. A group (Table S4, Figure S1b), 214 in C vs. A group (Table S5, Figure S1b), 47 in E vs. D group (Table S6, Figure S1b), and 141 in F vs. D group (Table S7, Figure S1b). These results showed that SCMs were abundantly enriched under freezing stress in both the species.

A total of 115, 189, 38, and 118 SCMs in different comparisons (B vs. A, C vs. A, E vs. D, and F vs. D, respectively) were assigned to the HMDB database (Figure 5). Ten classes were observed in the pie graph, and classification analysis revealed that "organooxygen compound", "prenol lipids", "flavonoids", and "fatty acyls" were the main metabolites in both CW and CO when exposed to cold stress (Figure 5). The class of "organooxygen compounds" contained the most active metabolites in CW and CO in the chilling stress treatment (B vs. A and E vs. D groups), and included 23 and 9 SCMs, respectively. The class of "prenol lipids" had the most active metabolites in CW and CO in the freezing stress treatment (C vs. A and F vs. D groups), including 42 and 23 SCMs, respectively. The number of "steroids and steroid derivatives" was relatively small in B vs. A and E vs. D groups, but was obviously increased in C vs. A and F vs. D groups (Figure 5).

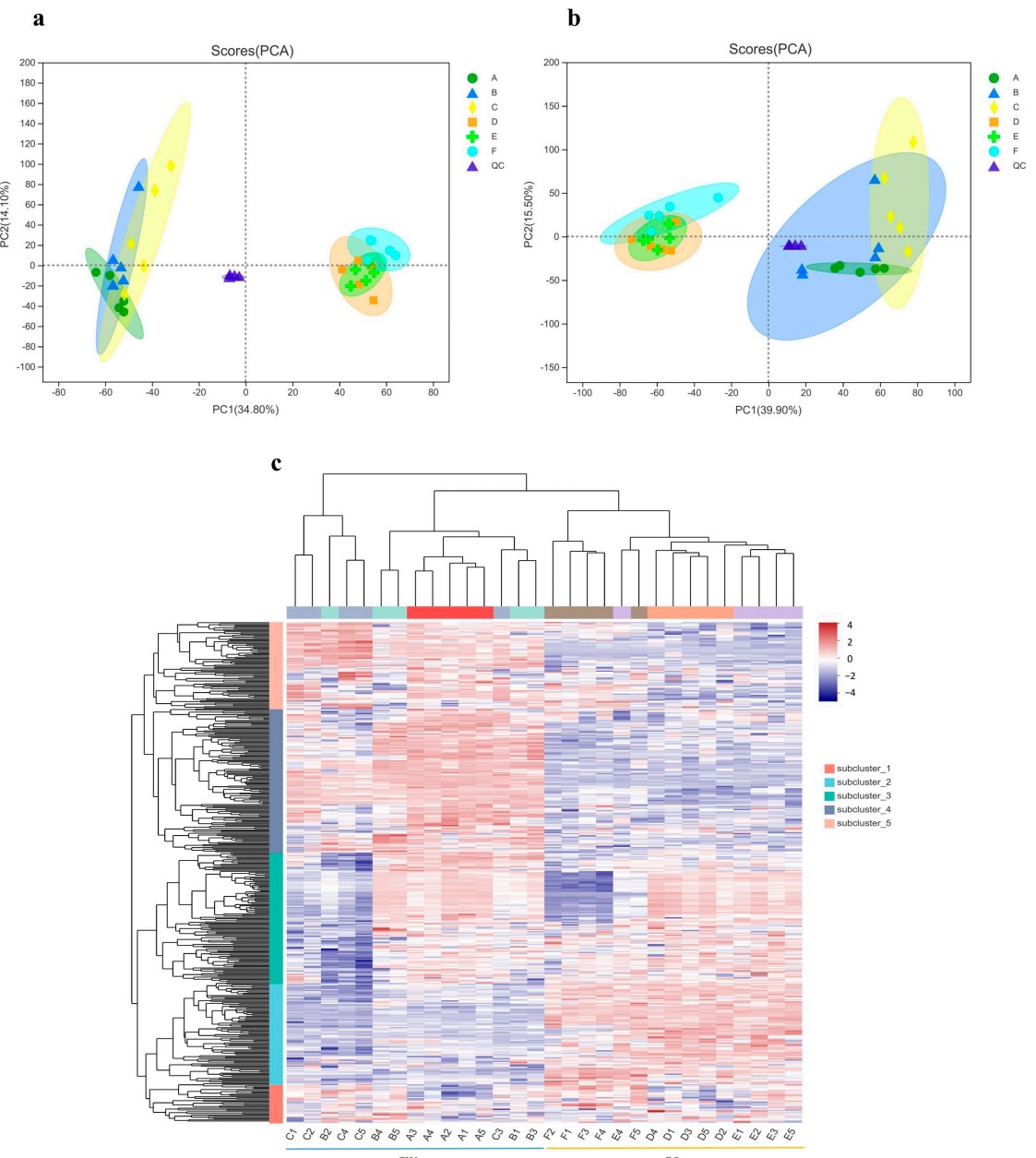

**Figure 4.** Principal component analysis (PCA) and hierarchical cluster analysis (HCA) of the metabolites distribution among different treatments. (**a**) PCA plot of positive mode, (**b**) PCA plot of negative mode, (**c**) clustering heat map of all metabolites in all samples by HCA, different colors are the values obtained after normalization of the metabolite content, red bars represent high abundance, whereas blue bars are low abundance. Five repetitions were taken for per treatment. CW, *Camellia weiningensis*; CO, *Camellia oleifera*; A, CW treated with 20 °C; B, CW treated with 4 °C; C, CW treated with −4 °C; D, CO treated with 20 °C; E, CO treated with 4 °C; F, CO treated with −4 °C. Quality control (QC) sample was the mixture of all samples.

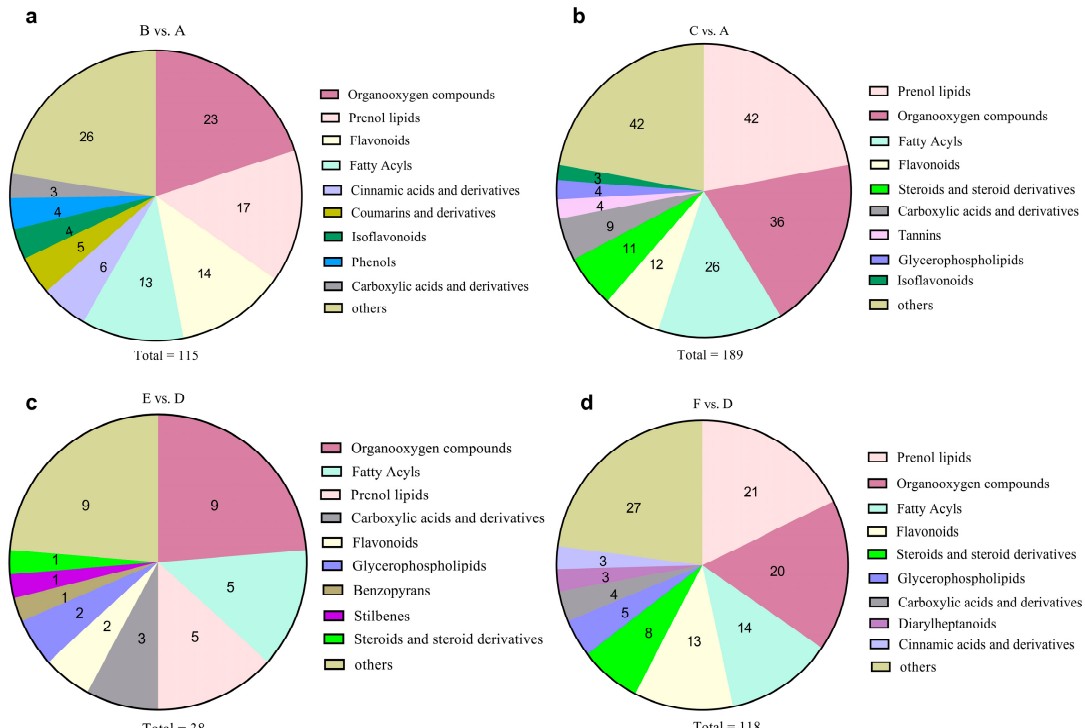

**Figure 5.** Classification of annotated SCMs using the HMDB database between different comparison groups, (**a**) B vs. A, (**b**) C vs. A, (**c**) E vs. D, (**d**) F vs. D. A, *C. weiningensis* treated with 20 °C; B, *C. weiningensis* treated with 4 °C; C, *C. weiningensis* treated with −4 °C; D, *C. oleifera* treated with 20 °C; E, *C. oleifera* treated with 4 °C; F, *C. oleifera* treated with −4 °C.

The top 30 SCMs according to the VIP value are shown in Figures 6 and 7. In CW seedlings under chilling stress (B vs. A), there were many SCMs with VIP value > 2.5; these included 8-pentanoylneosolaniol and ichangin 4-glucoside (classified as "prenol lipids"), ketotifen-*N*-glucuronide and neg_7671 (classified as "organooxygen compounds"), 5-L-glutamyl-L-alanine (classified as "carboxylic acids and derivatives") (Figure 6a). Canavalioside, ganoderic acid F, medicagenic acid (classified as "prenol lipids") quercetin-3′-glucuronide (classified as "flavonoids"), 1-O-caffeoyl-(b-D-glucose 6-O-sulfate) (classified as "cinnamic acids and derivatives"), and pos_7786 (classified as "isocoumarins and derivatives"), were the important metabolites in CO seedlings under chilling stress (Figure 7a). In CW and CO exposed to freezing stress (Figures 6b and 7b), most of the top 30 SCMs were downregulated, and were present at lower levels than in those exposed to the 20 °C treatment. In CW seedlings exposed to the −4 °C treatment, medicoside G and cynarasaponin F (classified as "prenol lipids"), yuccoside C and pos_5939 (classified as "steroids and steroid derivatives"), methionyl-proline (classified as "carboxylic acids and derivatives") and prostaglandin PGE2 1-glyceryl ester (classified as "fatty acyls"), play important roles in the resistance to freezing stress (Figure 6b). In CO seedlings exposed to the −4 °C treatment, there were many SCMs with VIP value > 3.5, such as steviobioside, medicoside G, and cynarasaponin F (classified as "prenol lipids"), yuccoside C, pos_5939, and digitoxigenin (classified as "steroids and steroid derivatives"), methionyl-proline (classified as "carboxylic acids and derivatives"), capsianoside I and 6-epi-7-isocucurbic acid glucoside (classified as "fatty acyls") (Figure 7b).

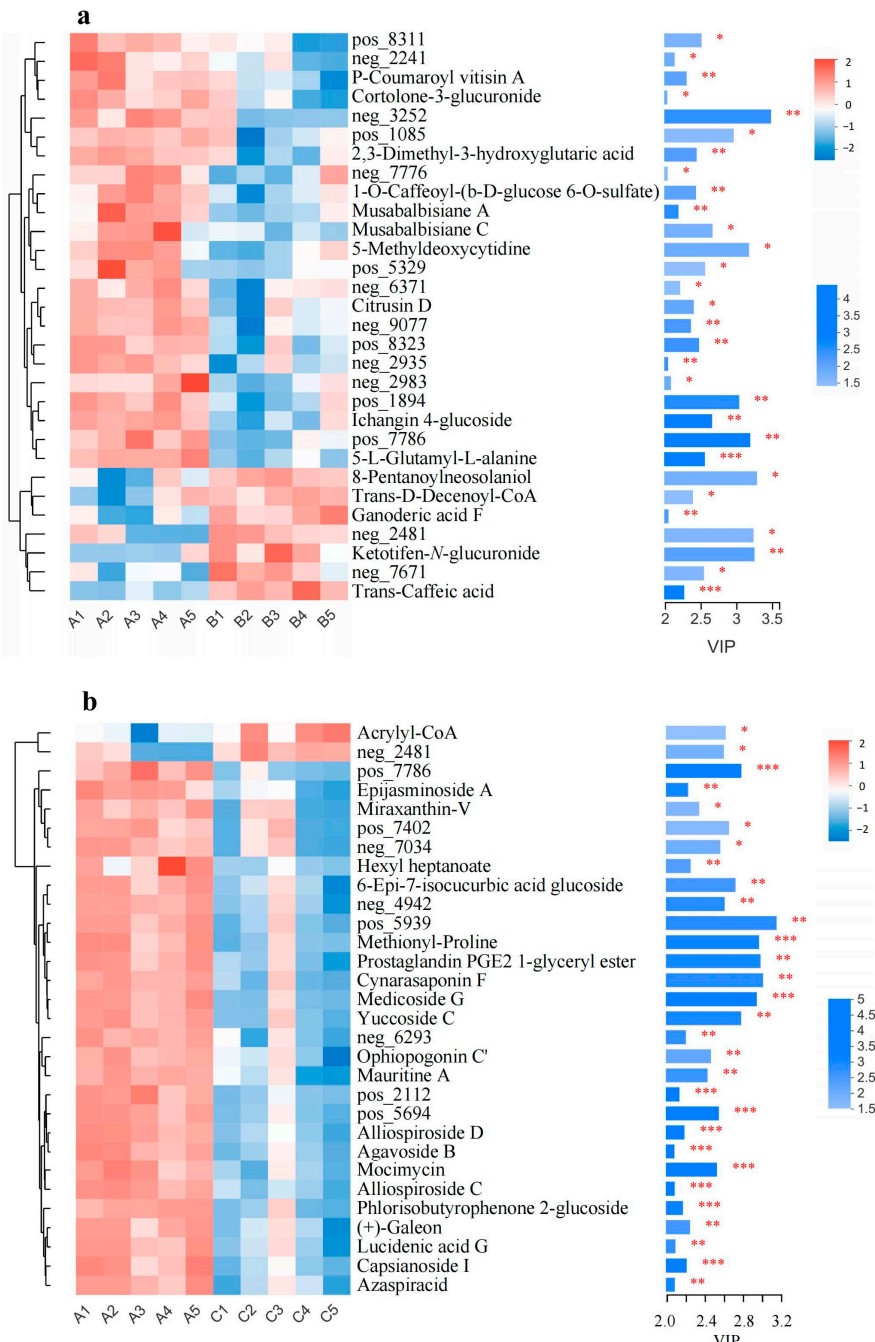

**Figure 6.** The top 30 SCMS with VIP value > 1 and *p* < 0.05 in CW seedlings under cold stress, (**a**) B vs. A (**b**) C vs. A. Clustering heat map is on the left, each column represents a sample, and the sample name is below. Each line represents a SCM, and the color represents the relative abundance of the metabolite in the samples. Red and blue colors mean higher and lower abundance, respectively. The VIP bar chart of the SCMs is on the right, the bar length indicates the contribution value of the metabolite to the difference between the two treatments, the greater the value, the greater the difference between the two treatments. The bar color indicates the significant difference (*p*-Value) of metabolites between the two treatments, the smaller the value, the larger the −log10 (*p*-Value) and the darker the color. Asterisk (*) indicates *p* < 0.05, asterisk (**) indicates *p* < 0.01, asterisk (***) indicates *p* < 0.001. A, *C. weiningensis* treated with 20 °C; B, *C. weiningensis* treated with 4 °C; C, *C. weiningensis* treated with −4 °C.

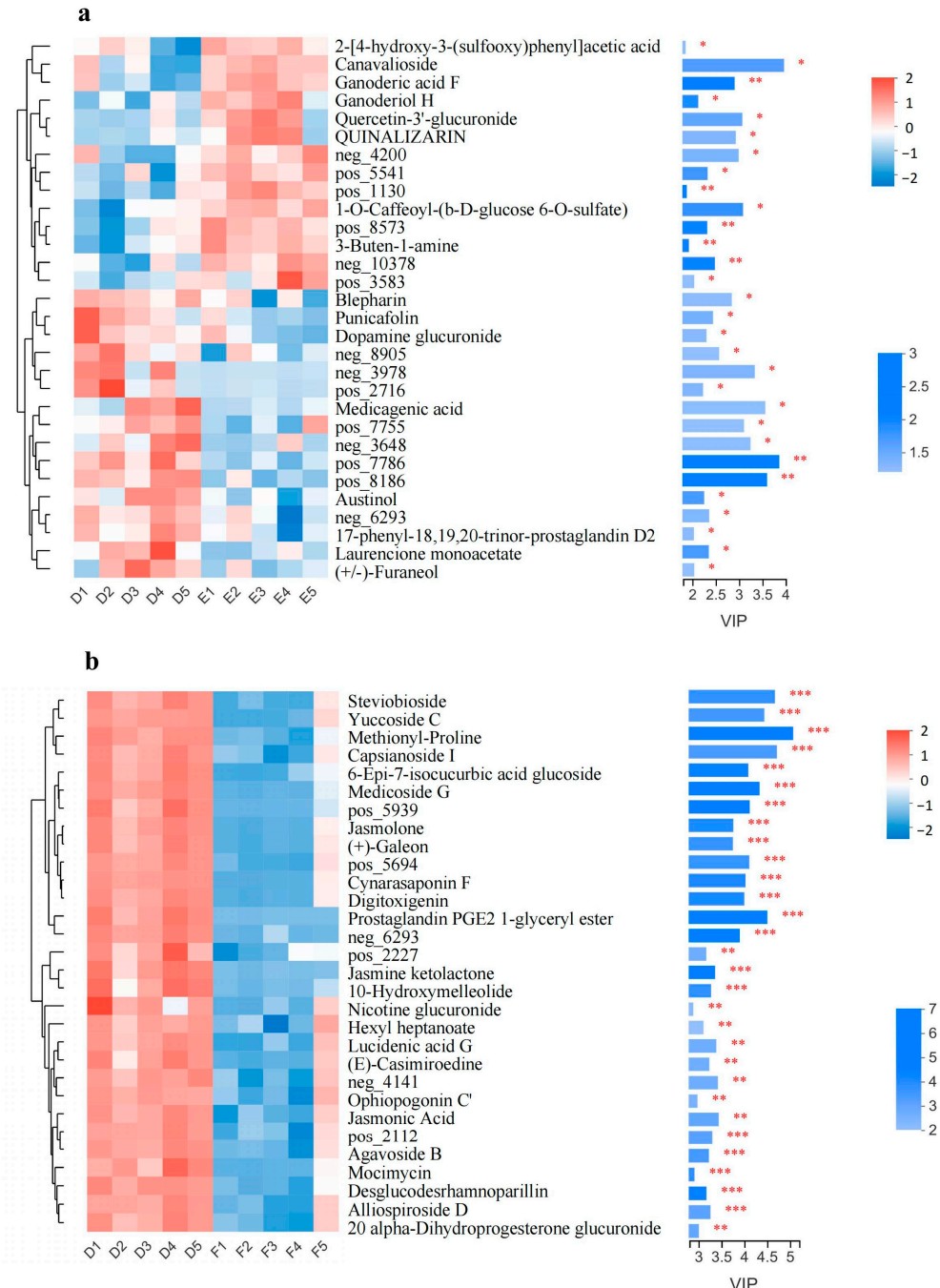

**Figure 7.** The top 30 SCMS with VIP value > 1 and *p* < 0.05 in CO seedlings under cold stress, (**a**) E vs. D (**b**) F vs. D. The VIP bar chart of the SCMs is on the right. The bar color indicates the significant difference (*p*-Value) of metabolites between the two treatments, the smaller the value, the larger the −log10 (*p*-Value) and the darker the color. Asterisk (*) indicates *p* < 0.05, asterisk (**) indicates *p* < 0.01, asterisk (***) indicates *p* < 0.001. D, *C. oleifera* treated with 20 °C; E, *C. oleifera* treated with 4 °C; F, *C. oleifera* treated with −4 °C.

### 3.6. Metabolic Pathway Analysis of Metabolites in CW and CO Seedlings

There were 388 SCMs between the two species under different temperature conditions, including 169 upregulated and 219 downregulated ones (Figure S2). Several key metabolic pathways associated with sugar metabolism, such as starch and sucrose metabolism, galactose metabolism, and amino sugar and nucleotide sugar metabolism, were found (Figure 8a). The key metabolites participated in sugar metabolism, such as UDP-glucose, UDP-D-apiose,

pseudaminic acid, fructose 6-phosphate, and UDP-*N*-acetylmuraminate, were present in higher contents in CW than in CO (Figure 8b).

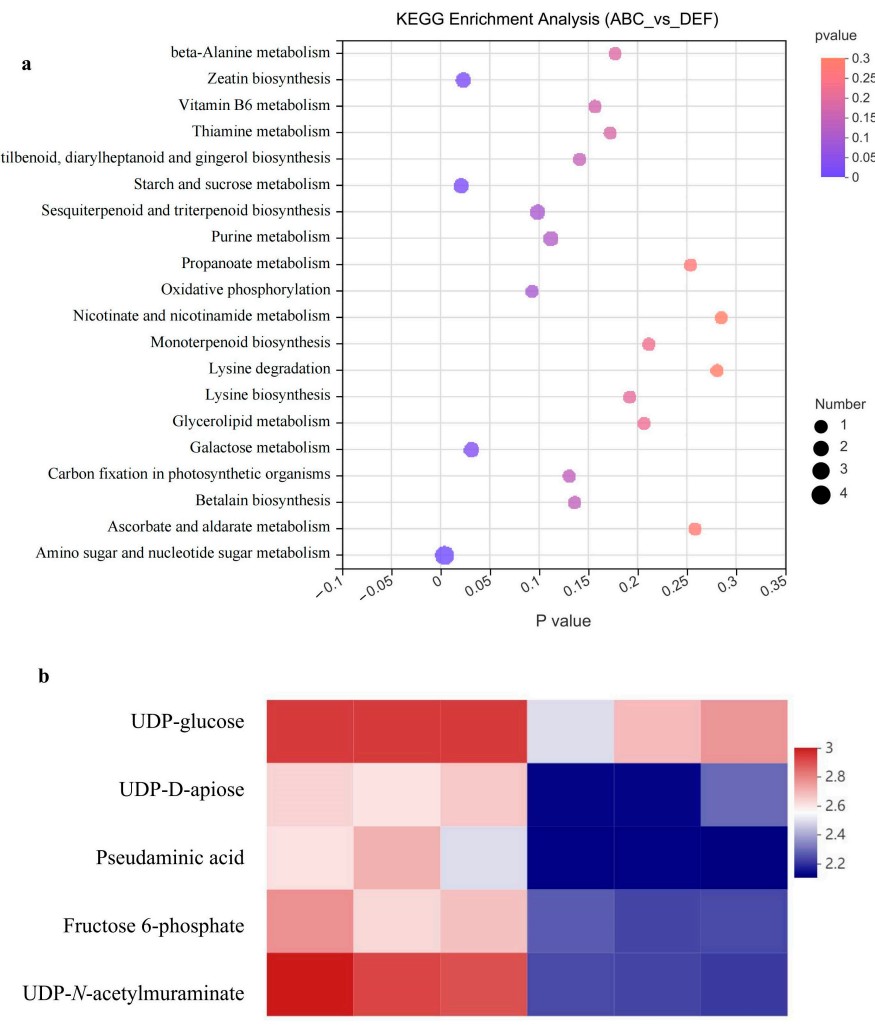

**Figure 8.** Metabolic pathway analysis of metabolites between two *Camellia* species. (**a**) KEGG pathway enrichment analysis of SCMs between CW and CO under different temperatures. The sizes of the dots represent the number of SCMs included in each row (KEGG pathway), and *p* values were calculated from hypergeometric tests; (**b**) heat map of key SCMs associated with sugar metabolism. The heat map was generated from the log2 (abundance values). Changes in abundance level are represented by a change in color; blue indicates a lower abundance level, whereas red indicates a higher abundance level. CW, *Camellia weiningensis*; CO, *Camellia oleifera*; A, CW treated with 20 °C; B, CW treated with 4 °C; C, CW treated with −4 °C; D, CO treated with 20 °C; E, CO treated with 4 °C; F, CO treated with −4 °C.

## 4. Discussion

*Camellia weiningensis* is suitable for growth in high altitude and low temperature environment found in the karst area in Guizhou. Changes in the anatomy of leaves in different *Camellia* species may be closely associated with their habitat. At high altitude, some plants have larger and thinner leaves than at lower altitude [26]. In this study, we found that the leaves of alpine CW seedlings were thinner than those of CO under normal conditions. Plants easily change the structure of leaves in response to low-temperature environment to enhance cold resistance [27]. The cold tolerance of plants correlates positively with leaf structure tightness and palisade-spongy tissue ratio [5]. Zeng et al. [8] report that

*C. oleifera* (Dabieshan No. 1), which has the highest tissue structure compactness, possesses the strongest cold resistance among six different CO cultivars. We found that CW seedlings had increased leaf and palisade thicknesses, and enhanced tightness of tissue structure and palisade-spongy tissue ratio in response to chilling stress (4 °C), and the results were similar to those reported by Wu et al. [9]. These results suggested that CW could change its leaf structure to enhance chilling stress tolerance. However, freezing stress (−4 °C) caused loosening of the leaf tissue structure and decreased the leaf tightness both in CW and CO, indicating that freezing stress caused obvious injury to the leaves of the two *Camellia* species at the seedling stage. Short-term cold stress may cause the phase change of cell membrane, make the cell membrane shrink [28], meanwhile cold stress are easy to change the intracellular osmotic substances to affect water loss, leading to affect the thickness of leaf tissue [28]. In this study, freezing stress may have a great impact on the second layer cells of palisade mesophyll, and the second layer cell structure became looser and more mixed with sponge tissue.

Low temperature not only affects the leaf structure, but also the photosynthesis. $F_v/F_m$ is an important chlorophyll fluorescence parameter that indicates the PS II activity when plants suffer from cold stress [29]. Low temperature leads to a reduction in the $F_v/F_m$ ratio. Gao et al. [11] report that PS II photoinhibition occurs in the leaves of CO under sub-chilling stress (10 °C/5 °C). Cold stress (5 °C) significantly decreases $F_v/F_m$ value in nine cultivars of *Stevia rebaudiana* [30]. In this study, the $F_v/F_m$ ratio were decreased significantly in CW and CO under freezing stress, but chilling stress had no significant change in CW when exposed to 4 °C, indicating that freezing stress reduces chlorophyll fluorescence in the two *Camellia* species, but CW may have strong tolerance to chilling stress.

Photosynthesis is critical for growth, development, and yield in plants, and is a sensitive physiological process when suffering from abiotic stresses [31]. Drought stress significantly reduces photosynthetic and chlorophyll fluorescence parameters, and maintain more intact mesophyll cell structures in Tree Peony (*Paeonia* section *Moutan* DC.), which is used for seed oil production in China [32]. In this study, we found that photosynthesis was significantly reduced both in CW and CO. The results showed similarly to the findings of Li et al. [33], Wu et al. [3,9], and Gao et al. [11]. Moreover, cold stress caused a reduction in the chlorophyll contents in the two *Camellia* species. Chlorophyll has an key function in the absorption and transport of light energy during photosynthesis, and the formation and stability of light-capturing chlorophyll a/b protein were necessary for photosynthesis [34,35]. A reduction in the chlorophyll content leads to low levels of light absorption, which lowers the photosynthetic rate and PS II efficiency [36]. In previous studies, we find that most of the genes encoding chlorophyll binding proteins and participating in the PS II are downregulated in CW seedlings under cold stress [13]. These results suggested that cold stress (especially freezing stress) caused major damage to the photosynthetic system in the two *Camellia* species at the seedling stage. Further studies are important to strengthen the prevention of freezing injury at the seedling stage in tea-oil tree.

Phytohormones are important growth regulators, have a major part in modulating the responses of plants to abiotic stress. *Camellia oleifera* GWu-2 improves ABA, IAA, and MeJA (methyl jasmonate) contents under drought stress, which may lead to stomata closing and decrease in water transpiration [37]. ABA, as a signaling agent, can improve tolerance to cold stress by regulating stomatal aperture and decreasing water loss [38]. Oliver et al. [39] report that cold stress induces an increase in endogenous ABA accumulation in rice. Ding et al. [40] found that drought stress increases the ABA levels in CO seedlings. These results are similar to our findings that ABA levels improved under 4 °C and −4 °C in the two *Camellia* species, suggesting that ABA plays a positive role in regulating cold stress tolerance. Garbero et al. [41] report that cold treatment reduces the IAA content in cold-sensitive *Digitaria eriantha* by 60% compared to that in control; however, in striking contrast, the IAA level in cold-resistant *D. eriantha* is 160% higher than in control at 24 h of cold treatment. We found that the IAA content increased significantly in CW seedlings under cold stress, but was reduced in CO seedlings, indicating that CW might

have improved cold resistance because of higher IAA content. JA is reported to positively regulate the ICE (inducer of CBF expression)–CBF (C-repeat binding factor) pathway to strengthen freezing resistance in *Arabidopsis* [42]. However, we found that the JA content was significantly decreased in CW and CO seedlings after cold treatment, indicating that JA might play a negative role in modulating low temperature tolerance. There is an interaction between hormones to resist low temperature, and further studies are needed to determine the functions of various hormones in different *Camellia* species.

Various abiotic stresses easily affect plant metabolism. Drought stress improves the contents of carbohydrates, amino acids, and some organic acids in *C. oleifera* "Changlin53" [14]. Heat stress causes a decrease in seed oil accumulation, and inhibits photosynthesis and respiration rates in *Brassica napus* seed [43]. In this study, PCA revealed a significant difference in metabolite composition between CO and CW, suggesting that the two *Camellia* species had evolved different metabolic mechanisms in response to cold stress. The habitats of the two species are different, especially with regard to the temperature requirements. The optimal growth temperature for CO is 16–18 °C, which is higher than that for CW [3,44], and it inevitably leads to considerable differences in their metabolism. Moreover, we found that freezing stress induced more metabolites, which may help *Camellia* species reduce the freezing damage. A variety of metabolites, such as prenol lipids, organooxygen compounds, and fatty acyls, were involved in the cold resistance of CO and CW. Prenol lipids mostly participate in the membrane structure, signal transduction, and energy storage [45,46], which have essential functions of defending against various abiotic stresses and pathogen infections [47]. Polyprenols help mangroves against salty seawater [48]. The synthesis of polyprenols in leaves of *Tilia × euchlora* trees mitigates salt stress [49]. Among the top 30 SCMs, most of the key metabolites, such as 8-pentanoylneosolaniol, canavalioside, and ganoderic acid F, belong to "prenol lipids", and their content was higher under chilling stress than under the control conditions, suggesting that CO and CW develop a defensive strategy by accumulating these polyprenols in leaves to alleviate chilling stress. Fatty acyls are the major building blocks of lipids and the most fundamental category of lipids in the biological system [50]. Cold can induce unsaturated fatty acid composition changes in drupes of the olive tree [51,52]. In a previous research, CW is found to differentially induce the expression of some genes related to fatty acid biosynthesis and degradation under cold stress [13]. In this study, 39 and 19 SCMs were classified as "fatty acyls" in CW and CO under cold stress, respectively; the VIP values for several of these were >2.5. These results indicate that fatty acid metabolism plays important roles in the response of CO and CW against cold stress. However, under freezing stress, the contents of most of the top 30 SCMs, such as medicoside G, cynarasaponin F, yuccoside C, and methionyl-proline, were lower than in the control both in CO and CW. Proline is a major osmolyte that maintains osmotic homeostasis and helps plants guard against various abiotic stress. Our results are contrary to those of Khalid et al. [53,54], who found that grafted Volkamer lemons had significantly higher proline content under different water-deficit regimes and salt conditions. The results indicated that the proline metabolic pathway might be repressed in CO and CW under freezing stresses.

Low temperature affects plant sugar metabolism. For example, maize accumulates a large number of sugar metabolites, such as raffinose, trehalose-6-phosphate, fructose, fructose-6-phosphate, and glucose to cope with the combined stress of low temperature and drought [55]. Under 6 °C treatment, the activities of trehalose-6-phosphate synthase and trehalose phosphate phosphatase in sugarcane are significantly increased, and the trehalose content is significantly increased [56]. In this study, we found that the contents of some key metabolites associated with sugar metabolism, such as UDP-glucose, UDP-D-apiose, and fructose 6-phosphate, were higher in CW than in CO. These results suggested that sugar metabolism might contribute to enhanced cold resistance of CW; however, further studies are needed to verify the special functions of sugar metabolism in CW.

## 5. Conclusions

In this study, CW seedlings exhibited increased leaf thicknesses and tissue structure tightness to enhance chilling stress (4 °C) tolerance, but freezing stress (−4 °C) caused loosening of the leaf tissue structure in CW. CO seedling reduced leaf, palisade, and spongy tissue thicknesses both in 4 °C and −4 °C conditions. Photosynthesis was inhibited in the two *Camellia* species under cold stress. ABA levels were improved in CW and CO under cold stress, but JA levels were decreased. CW seedlings also increased IAA levels as the decreasing temperatures. Chilling and freezing stress induced more SCMs in CW compared to that in CO. Prenol lipids, organooxygen compounds, and fatty acyls were the main metabolites that participated in the response of the two *Camellia* species to cold stress. Sugar metabolite-related pathways, such as starch and sucrose metabolism, galactose metabolism, and amino sugar and nucleotide sugar metabolism were significantly enriched in CW, and the key SCMs, such as UDP-glucose and fructose 6-phosphate had higher contents in CW than in CO. This study shows the differences and similarities in the mechanisms of cold resistance in the two *Camellia* species, and contributes to improving cold tolerance in *Camellia*, and shows that new strategies should be developed to improving the breeding of *Camellia* in alpine and cold areas.

**Supplementary Materials:** The following supporting information can be downloaded at: https://www.mdpi.com/article/10.3390/horticulturae8060494/s1, Figure S1: (a) Venn diagram showing all SCMs numbers among different comparison groups; Figure S2: Volcano plot of SCMs between two *Camellia* species seedlings; Table S1: Two-way ANOVA analysis the effect of temperature and and species on leaf anatomical structure; Table S2: Two-way ANOVA analysis the effect of temperature and and species on photosynthesis; Table S3: Two-way ANOVA analysis the effect of temperature and species on endogenous hormone contents; Table S4: The SCMS in B vs. A group; Table S5: The SCMS in C vs. A group; Table S6: The SCMS in E vs D group; Table S7: The SCMS in F vs D group.

**Author Contributions:** H.X.: conceptualization, investigation, writing—original draft, funding acquisition. C.H.: methodology, supervision. X.J. and J.Z.: performed the experiments. X.G.: resources, funding acquisition. C.Y.: methodology, resources, writing—review and editing, funding acquisition. All authors have read and agreed to the published version of the manuscript.

**Funding:** This work was supported by the science and technology support projects of Guizhou Province (Qian ke he foundation ZK [2022] General 210), Guizhou Education Cooperation (Qian jiao he KY Word [2021]106), and Guizhou minzu university Natural Science Foundation (GZMU [2019]YB17 and GZMUZK [2021]YB17).

**Institutional Review Board Statement:** Not applicable.

**Informed Consent Statement:** Not applicable.

**Data Availability Statement:** Data are contained within the article.

**Conflicts of Interest:** The authors declare no conflict of interest.

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
