# Peer review of "Impact of Cold Stress on Leaf Structure, Photosynthesis, and Metabolites in Camellia weiningensis and C. oleifera Seedlings"

_horticulturae, doi:10.3390/horticulturae8060494_

Round 1

Reviewer 1 Report

The manuscript by Xu et al. investigated the effects of cold stress on leaf structure, photosynthetic efficiency, and various metabolic compounds in two different Camellia species and have shown that Camellia weiningensis seedlings exhibitbetter effectivity in response to cold stress. Although, the manuscript was very well written and the authors performed in-depth comparison of various physiological, morphological characteristics and metabolic profile of two different Camellia species to show that Camellia weiningensis is suitable for growing in mountainous areas with relatively lower temperature and can be used for cultivation in those areas, I have some comments which I believe will help authors to improve their manuscript before resubmission.  

  • L19: Cold stress increased ABA content in both Camellia species, but which one exhibit the highest amount? Please clearly mention that here
  • L24-25: The top key SCMs, …..freezing stress in which Camellia species? Please mention in the revised manuscript
  • Is the only aim of the study is to compare various parameters in two Camellia species and suggest the suitability of one of them in cold and mountainous climate? I think, this study can provide other important take homes. If possible, please highlight them with possible application in a sentence or two in the abstract to gain attention of bigger group of readers.
  • Introduction should be improved with more relevant information and their citations: for example, cite some excellent papers that show the effects of altitude and cold stress on Fatty acids and phytohormones. Last paragraph should also be improved by adding one or two sentences indicating the key findings.
  • L85: Camellia oleifera (Chang Lin) and weiningensis seedlings or what?
  • L98: What is the exact composition of formaldehyde alcohol acetic acid solution?
  • L104-107: Please cite reference(s) for all these or explain more
  • L112-113: Please provide citation(s)

Material and methods: 2.5-Quantification of plant hormones: This part has been written very poorly. Please improve this section by providing more details. Also, for phytochemicals/phytohormones HPLC-Based or GC-Based detection are more common and more efficiently used than immunoassays.

Results: 3.1: Changes in anatomical features of leaves in the two Camellia species under cold stress--- Are all the comparisons qualitative through visualization or any quantitative data available from image analysis?

3.2-3.4: The results from these sections are interesting, but could be improved, so they can be more easily assessed by the readers. First of all, I see no statistics anywhere, aside from some posthoc letters in some panels. As a reader, I would like to know the stats for effects of species and temperature etc.

L323: ICE–CBF pathway—Full form please

Also, the discussion can still be improved by adding more relevant references of effects of temperature or other abiotic stressors on morpho-physiological and biochemical changes in other oil seed/oil tea species

Finally: Conclusion must be improved. At present, there are redundancies of this part with other sections. The conclusion should unequivocally explain the key findings in more details and some of the future applications and directions of research in this area. Please improve this section.

Author Response

Dear Reviewer,

Thank you for your good work on our manuscript, and we are very grateful to Reviewer’s suggestion. Based on these comments and suggestions, we have carefully revised our original manuscript. We also responded point by point to each reviewer’s comments, along with a clear indication of the location of the revision. Hope these will make it more acceptable for publication.

Response to review 1#

L19: Cold stress increased ABA content in both Camellia species, but which one exhibit the highest amount? Please clearly mention that here

Response: According to the suggestion, we had added it (line 19-20).

L24-25: The top key SCMs, …..freezing stress in which Camellia species? Please mention in the revised manuscript

Response:According to the suggestion, we had revised it (line 26).

Is the only aim of the study is to compare various parameters in two Camellia species and suggest the suitability of one of them in cold and mountainous climate? I think, this study can provide other important take homes. If possible, please highlight them with possible application in a sentence or two in the abstract to gain attention of bigger group of readers.

Response: we had added highlighted the possible application in the abstract (line 29-31).

Introduction should be improved with more relevant information and their citations: for example, cite some excellent papers that show the effects of altitude and cold stress on Fatty acids and phytohormones. Last paragraph should also be improved by adding one or two sentences indicating the key findings.

Response:We had added some studies that showed the effects of altitude and cold stress on fatty acids and phytohormones (line 74-79). We also added the key findings in the last paragraph (line 89-91).

L85: Camellia oleifera (Chang Lin) and weiningensis seedlings or what?

Response:here it refers to “seedlings”, we had added it (line 96).

L98: What is the exact composition of formaldehyde alcohol acetic acid solution?

Response:we had added the exact composition “38% formaldehyde, 70% alcohol, and acetic acid solution (5:90:5,v/v/v)” (line 111-112).

L104-107: Please cite reference(s) for all these or explain more

Response:we had cited related reference (line 117). 

L112-113: Please provide citation(s)

Response:According to the suggestion, we had provided the related citation (line 120).

Material and methods: 2.5-Quantification of plant hormones: This part has been written very poorly. Please improve this section by providing more details. Also, for phytochemicals/phytohormones HPLC-Based or GC-Based detection are more common and more efficiently used than immunoassays.

Response:According to the suggestion, we had improved this sections through more detailed description (line 142-152). Yes, we will improve the experimental method for phytohormones measurement by HPLC-Based or GC-Based in the future experiments.   

Results: 3.1: Changes in anatomical features of leaves in the two Camellia species under cold stress--- Are all the comparisons qualitative through visualization or any quantitative data available from image analysis?

Response:All the comparisons were quantitative data available from image analysis by the Image J software. We had added it (line 210-211 ). The thicknesses of leaf, upper and lower epidermis, and spongy and palisade tissues were measured using the Image J software (version 1.8.0).

3.2-3.4:The results from these sections are interesting, but could be improved, so they can be more easily assessed by the readers. First of all, I see no statistics anywhere, aside from some posthoc letters in some panels. As a reader, I would like to know the stats for effects of species and temperature etc.

Response:According to the suggestion, we had added the Statistical analysis in the methods (line 200-206). We also added the two factor analysis of variance to show the effects of species and temperature (line 219-221, 242-245, 261-265, Table S1-S3).

 the stats for effects of species and temperature

L323: ICE–CBF pathway—Full form please

Response:we had added the full form of “ICE (inducer of CBF expression)–CBF (C-repeat binding factor) pathway ” (line 434-435)

Also, the discussion can still be improved by adding more relevant references of effects of temperature or other abiotic stressors on morpho-physiological and biochemical changes in other oil seed/oil tea species

Response:We had improved the discussion and added some references of effects of abiotic stressors on morpho-physiological and biochemical changes in other oil seed/oil tea species (line 402-405, 420-422, 441-445, 463-464).

Finally: Conclusion must be improved. At present, there are redundancies of this part with other sections. The conclusion should unequivocally explain the key findings in more details and some of the future applications and directions of research in this area. Please improve this section.

Response:we had rewrited the Conclusion, we had explained the key findings in more details and added some of the future applications (line 490-506).

Reviewer 2 Report

The presented manuscript compares two Camellia species that differ in their susceptibility to low temperatures. The work is interesting, and performed experiments are of high quality. However, I have several important questions and comments that should be addressed before the publication of this manuscript.

Major comments:

  1. l: 93-94  - can the Authors specify how long the low-temperature stress was applied? From the description, I understood that it was for 24 h only. If so, can the Authors explain how such short-term stress could influence, e.g., the thickness of the leaf blade, which develops for a substantially longer time? For instance, exemplary images for CO show a different number of palisade mesophyll layers in control and low-temperature treated plants. Such variability can be introduced during blade development but not due to short-term stress applied to mature leaves.
  2. l: 115 - Please explain why Chl content and Chl in vivo fluorescence were measured after 1-day recovery from cold stress?
  3. Figure 1 - what is the N number for performed measurements and from how many images the measurements were performed?; note that information about the repetitions of performed experiments is also missing for results presented in all other Figures.
  4. l: 187 and further in the text (Fv/Fm parameter) - due to the latest analyses regarding the interpretation of the Fv/Fm  https://doi.org/10.1093/plcell/koab008; please provide the current understanding of this parameter which cannot be equated with the quantum efficiency of PSII photochemistry.

Minor comments:

  1. l: 44 'camellia family' I assume the Authors meant 'camellia genus'?
  2. l: 91 please provide light intensity (PAR value) in which plants were growing
  3. l:101-102 'fast green double dyes' - please specify those dies
  4. l: 129 - please specify what is the leaf powder?
  5. l: 301-303 - sentence starting with 'Chlorophyll captures...' is unclear; please rewrite it.
  6. the text requires further spell and grammar checks - e.g., 'can reaches' (l:44)

Author Response

Dear Reviewer,

Thank you for your good work on our manuscript, and we are very grateful to the Reviewer’s suggestion. Based on these comments and suggestions, we have carefully revised our original manuscript. We also responded point by point to each reviewer’s comments, along with a clear indication of the location of the revision. Hope these will make it more acceptable for publication.

Response to review 2#

  1. l: 93-94  - can the Authors specify how long the low-temperature stress was applied? From the description, I understood that it was for 24 h only. If so, can the Authors explain how such short-term stress could influence, e.g., the thickness of the leaf blade, which develops for a substantially longer time? For instance, exemplary images for CO show a different number of palisade mesophyll layers in control and low-temperature treated plants. Such variability can be introduced during blade development but not due to short-term stress applied to mature leaves.

Response:We had added the explanation in the discussion (Line 385-390). 

“There are two layers of palisade mesophyll both in CW and CO under control and cold stress, and low temperature did not change the number of palisade mesophyll layers. But short-term stress may cause the phase change of cell membrane, make the cell membrane shrink (Yadav, 2010), meanwhile cold stress are easy to change the intracellular osmotic substances to affect water loss, leading to affect the thickness of leaf tissue (Yadav, 2010). In this study, freezing stress may have a great impact on the second layer cells of palisade mesophyll, and the second layer cell structure became more loose and mixed with sponge tissue.”

Reference:

Yadav, S.K. Cold stress tolerance mechanisms in plants. Agron. Sustain. Dev. 2010, 303: 515-527.

  1. l: 115 - Please explain why Chl content and Chl in vivofluorescence were measured after 1-day recovery from cold stress?

Response:Because photosynthetic parameters measured weakly after cold treatment, according to the methods (11, 27), to better assess the low temperature damage to photosynthetic system, photosynthetic parameters were measured after a period of time. In order to be consistent with photosynthetic parameters, Chl content and Chl fluorescence were also measured after 1-day recovery from cold stress.

Reference:

 Gao, L.; Wang, Q.; Chen, Y.; Sun, Y.; Zhang, L. Effects of Sub-chilling Stress in Spring on Photosynthetic Physiological Characteristics of Camellia Oleifera ‘Huashuo’ Leaves. Journal of Southwest forestry university, 2021, 41:1-8. (In chinese)

  1.  Li, X., Ahammed, G.J, Li, Z.X., Zhang, L., Wei, J.P., Yan, P., Zhang, L.P, Han, W.Y. Freezing stress deteriorates tea quality of new flush by inducing photosynthetic inhibition and oxidative stress in mature leaves. Sci. Hortic.2017, 230: 155-160.
  2. Figure 1 - what is the N number for performed measurements and from how many images the measurements were performed?; note that information about the repetitions of performed experiments is also missing for results presented in all other Figures.

Response:We had added the description of experiment repetition in figure legends 1-4.

“In Figure 1, six plants of each of the two species from cold treatment and control were used for the measurements, five images per plant were taken. ”

  1. l: 187 and further in the text (Fv/Fm parameter) - due to the latest analyses regarding the interpretation of the Fv/Fm  https://doi.org/10.1093/plcell/koab008; please provide the current understanding of this parameter which cannot be equated with the quantum efficiency of PSII photochemistry.

Response:yes, we had corrected the understanding of Fv/Fm parameter (line 234-236).

Minor comments:

  1. l: 44 'camellia family' I assume the Authors meant 'camellia genus'?

Response:We had revised it (line 43 ).

  1. l: 91 please provide light intensity (PAR value) in which plants were growing

Response:According to the suggestion, we had added the light intensity (line 103).

  1. l:101-102 'fast green double dyes' - please specify those dies

Response:We had specified the dyes, and we used two kind of dyes (Safranine and Fast green) according to Ruzin’s method (line 114-115).

  1. l: 129 - please specify what is the leaf powder?

Response:We had corrected it as follows: “leaves are ground into powder with liquid nitrogen, and leaf powder (0.1 g per sample) (line 137-138).” 

  1. l: 301-303 - sentence starting with 'Chlorophyll captures...' is unclear; please rewrite it.

Response:We had rewrited it (line 409-411).  

“Chlorophyll has an key function in the absorption and transport of light energy during photosynthesis, and the formation and stability of light-capturing chlorophyll a/b protein were necessary for photosynthesis”

  1. the text requires further spell and grammar checks - e.g., 'can reaches' (l:44)

Response:According to the suggestion, we had revised the spell and grammar carefully.

Round 2

Reviewer 1 Report

I think, the authors made substantial changes to improve the manuscript as suggested. The current form is ready for acceptance.  

Author Response

We have carefully revised the language and added some relevant references in the introduction.